# Simulated Replications as a Methodological Tool in Social Science Research

## Abstract

Simulation is increasingly recognized as a methodological complement to human-subjects research in the social sciences. This study demonstrates the potential and limitations of simulated data by replicating a published experiment on race cues in mediated communication. Using an AI-enabled workflow, we reproduced the design of Hong et al. [2024], which tested the effects of creator race and influencer race on evaluation, credibility, message acceptance, and engagement. A simulated panel of 240 participants was generated across four experimental conditions and a control group. Statistical analyses showed a partial replication: the main effect of creator race on credibility, a central finding in Hong et al.'s human-sample data, was reproduced and even amplified. Message acceptance again showed null effects. However, the effects of influencer race observed in Hong et al. [2024] were absent, while the effects on evaluation and participation were exaggerated. These results highlight both the promise and the pitfalls of simulation: strong effects may be recoverable, but subtle, context-dependent differences may be lost. More broadly, simulation offers a pathway for accelerating theory testing, replication, and methodological innovation in social science research.

## 1 Introduction

The communication sciences have long relied on experimental and survey methods to understand how audiences process messages and evaluate communicators. Yet the replication crisis across psychology and related disciplines has underscored the need for complementary approaches to testing theoretical claims [Camerer et al., 2018, Collaboration, 2015]. Simulation, long used in computational sciences, is now emerging as a methodological frontier in social science research [Epstein, 1999]. By generating artificial samples under controlled parameters, researchers can explore whether effects observed in human participants reappear in constrained artificial contexts, thereby clarifying which effects are robust and which are fragile.

Although rarely applied in communication, simulation offers at least three potential contributions. First, it can serve as a **replication tool**, testing whether previously reported effects can be reproduced under controlled artificial conditions. Second, it can serve as a **theory probe**, clarifying whether hypothesized effects emerge even when variability is reduced. Third, it can provide **early validation for experimental designs**, allowing researchers to refine hypotheses before committing to resource-intensive data collection [Park et al., 2023, Argyle et al., 2023].

To illustrate this approach, the present study conducted a simulation-based replication of Hong et al. [2024], who examined how racial cues influence perceptions of credibility, evaluation, and engagement in mediated communication [Hancock et al., 2020]. Their results showed that the racial identity of the *creator* of a message strongly predicted perceptions of credibility, whereas the race of the *communicator* (in their case, a virtual influencer [Kim and Wang, 2024]) yielded weaker and less consistent effects. Message acceptance and engagement were largely unaffected. The current

study reproduced this design using AI-driven survey construction and simulated participants, asking whether these findings would generalize to artificial data.

## 1.1  Research Objectives

The present research, therefore, seeks to answer two core questions: (1) Can simulated agent data replicate the main findings of Hong et al. [2024], particularly the primacy of creator race in shaping credibility perceptions? and (2) To what extent do simulations reproduce or diverge from subtle effects of influencer race and interaction effects between creator and influencer identities? In addressing these questions, the study contributes to methodological debates about the use of AI and simulated datasets in social science research.

## 1.2  Hypotheses and Research Questions in the Original Study

In the original study, Hong et al. [2024] advanced two central hypotheses that guided their experimental design. H1 predicted that the race of the creator would significantly shape audience perceptions. Specifically, messages attributed to a Black creator, compared to a White creator, were expected to yield more favorable evaluations of the communicator, greater message acceptance, higher perceived credibility of the creator, and stronger engagement intentions. The results provided partial support for this hypothesis. Among the four outcome measures, only credibility showed a significant effect: Black creators were rated as more credible than White creators. However, creator race did not significantly influence evaluation, message acceptance, or engagement intentions.

H2 proposed that the race of the communicator would exert similar effects on audience responses. That is, messages delivered by a Black communicator, compared to a White communicator, were hypothesized to enhance evaluation, message acceptance, credibility, and engagement. This hypothesis was not supported. Across all outcome measures, communicator race failed to produce significant differences.

These two hypotheses formed the focal point of the present simulation-based replication, as they directly addressed the experimental manipulations of creator and communicator race and their effects on audience perceptions.

## 2  Method

## 2.1  Research Design and Tools

This study employed a mixed-methods approach that integrated artificial intelligence (AI)–driven research tools with simulated panel data collection, utilizing Liner Research Agents (**https://getliner.com/**). Survey design and data generation were conducted by converting PDF-based surveys into interactive, programmable survey instruments with customizable participant constraints. The process unfolded in four structured stages.

In **Stage 1**, the original survey instrument, modeled on Hong et al. [2024], was uploaded in PDF format to the Panel Agent interface. The uploaded file contained measures of influencer evaluation, message acceptance, credibility, and engagement intentions, each assessed using established Likert-type scales. In **Stage 2**, the system parsed the uploaded survey and automatically extracted individual questions, which were then reviewed and edited for clarity. Researchers were able to add, modify, or remove content blocks, ensuring fidelity to the original instrument while preserving methodological flexibility. In addition to text-based research support, Liner AI is also capable of recognizing and processing images, allowing experimental stimuli to be embedded in studies not only as text but also as visual material.

In **Stage 3**, persona constraints were applied to simulate a target population. These included setting the number of panel participants, establishing an age range of 18–65 years, and applying a minimum education requirement of high school completion. Additional custom constraints, such as gender or occupation, could be included as needed. In **Stage 4**, the survey was deployed to the simulated participant pool. The system tracked progress, response quality, and dropout rates. The simulated data collection achieved a 0% dropout rate, ensuring complete datasets across all experimental conditions. Finally, in **Stage 5**, the Panel Agent generated an automated report summarizing survey results and response patterns. This included descriptive statistics, Likert-scale distributions, and AI-generated

interpretation of response patterns. These outputs were exported and integrated with the statistical analyses conducted in R.

## 2.2 Participants and Conditions

The simulated panel consisted of 200 participants, distributed evenly across the four factorial conditions: Black Creator–Black Influencer (BCBI), Black Creator–White Influencer (BCWI), White Creator–Black Influencer (WCBI), and White Creator–White Influencer (WCWI). An additional 40 participants formed a control group, producing a total of $N = 240$ observations. Each simulated participant was constrained by the predefined persona filters described above, which ensured representativeness across age and education criteria.

## 2.3 Measures

Dependent variables were adapted directly from Hong et al. [2024]:

- **Evaluation (Liking)** – Participants rated overall liking of the virtual influencer on a 7-point Likert scale.

- **Message Acceptance** – Agreement with and support for the influencer's message.

- **Credibility (Trustworthiness)** – Perceived expertise and trustworthiness of the creator.

- **Engagement Intentions** – Willingness to share, comment on, or further engage with the influencer's post.

## 2.4 Analytic Strategy

Data were analyzed in three stages. First, descriptive statistics were computed to establish baseline comparisons with Hong et al. [2024]. Second, two-way ANOVAs were conducted to examine the main and interaction effects of creator race and influencer race on each dependent variable. Finally, effect size comparisons were performed to evaluate whether the simulated dataset reproduced the magnitude and direction of Hong et al. [2024]'s findings.

# 3 Results

## 3.1 Descriptive statistics

Across the simulated sample, mean scores were as follows. For evaluation, participants reported moderate-to-high liking, with a grand mean of 4.30 and a standard deviation of 0.48. For message acceptance, scores were higher, with a grand mean of 4.85 and a standard deviation of 0.44. For credibility, ratings were also moderate to high, with a mean of 4.35 and a standard deviation of 0.52. For engagement, scores were moderate, with a mean of 3.90 and a standard deviation of 0.54. Compared to Hong et al. [2024], simulated data produced higher engagement scores and slightly higher evaluations, while credibility and message acceptance means were comparable.

## 3.2 ANOVA tests

A series of two-way ANOVAs examined the effects of creator race and communicator race on each dependent variable. For evaluation, the main effect of creator race was statistically significant, $F(1,156) = 42.94$, $p < .001$, partial $\eta_p^2 = .22$. Communicators paired with Black creators received higher evaluations ($M = 4.56$, $SD = 0.47$) than those paired with White creators ($M = 4.03$, $SD = 0.45$). The main effect of communicator race was not significant, $F(1,156) = 2.15$, $p = .144$, partial $\eta_p^2 = .01$, nor was the interaction, $F(1,156) = 0.36$, $p = .552$, partial $\eta_p^2 = .00$.

For credibility, the main effect of creator race was again significant, $F(1,156) = 52.62$, $p < .001$, partial $\eta_p^2 = .25$. Black creators were judged more credible ($M = 4.70$, $SD = 0.47$) than White creators ($M = 4.00$, $SD = 0.46$). No effects were found for communicator race, $F(1,156) = 0.05$, $p = .819$, partial $\eta_p^2 = .00$, or for the interaction, $F(1,156) = 0.02$, $p = .880$, partial $\eta_p^2 = .00$.

Table 1: replication summary comparing human and AI samples

| Dependent variable | Hong et al. (2024): human | Simulation replication: AI | Replication outcome |
|---|---|---|---|
| **H1. Creator race (Black vs. White)** | | | |
| Evaluation (liking) | n.s. | Significant, Black > White ($p < .001$) | Divergent (inflated) |
| Message acceptance | n.s. | n.s. | Convergent (null) |
| Credibility | Significant, Black > White ($p < .05$) | Significant, Black > White ($p < .001$) | Convergent (amplified) |
| Engagement intentions | n.s. | Significant, Black > White ($p < .05$) | Divergent (new) |
| **H2. Communicator race (Black vs. White)** | | | |
| Evaluation (liking) | n.s. / modest trend (Black > White) | n.s. | Divergent (disappeared) |
| Message acceptance | n.s. | n.s. | Convergent (null) |
| Credibility | n.s. / modest trend (Black > White) | n.s. | Divergent (disappeared) |
| Engagement intentions | n.s. | n.s. | Convergent (null) |

*Note*. n.s. = not significant. "Convergent" indicates directionally consistent replication; "Divergent" indicates a discrepancy (inflated/new/disappeared).

For engagement, results showed a smaller but statistically significant main effect of creator race, $F(1,156) = 4.38$, $p = .038$, partial $\eta_p^2 = .03$. Black creators elicited greater engagement intentions ($M = 4.02$, $SD = 0.52$) compared to White creators ($M = 3.78$, $SD = 0.51$). The main effect of communicator race was nonsignificant, $F(1,156) = 0.84$, $p = .361$, partial $\eta_p^2 = .01$, as was the interaction, $F(1,156) = 0.12$, $p = .731$, partial $\eta_p^2 = .00$.

For message acceptance, neither creator race nor communicator race yielded significant main effects, all $ps > .10$. **Table 1** compares the hypothesis tests in Hong et al. [2024]'s original experiment with the simulation outcomes.

## 3.3 Effect size comparisons

Comparisons with Hong et al. [2024] indicate both convergence and divergence. The replicated finding of a main effect of creator race on credibility was consistent across both studies, though the effect was larger in the simulated dataset (mean difference = 0.70) compared to Hong's original (= 0.35). The positive effect of creator race on evaluation was also stronger in the simulation (mean difference = 0.53) compared to Hong (0.37). Engagement showed a small but significant effect in the simulation (mean difference = 0.26), whereas Hong's study reported null results. By contrast, communicator race effects were negligible in the simulation: mean differences were 0.12 for evaluation and –0.01 for credibility, compared to Hong's reported 0.26 and 0.35, respectively.

## 4 Discussion

The purpose of this study was to examine whether simulated datasets could reproduce previously reported findings in communication research. Using Hong et al. [2024] as a test case, results demonstrate a **partial replication**. The robust effect of creator race on credibility was reproduced, and the null effect of message acceptance was replicated. However, the simulation exaggerated creator-race effects on evaluation and engagement while failing to capture communicator-race effects observed in the original.

These outcomes carry several implications. First, they suggest that simulations are most successful at recovering **large, robust effects** that align with strong theoretical predictions. The consistency of the creator-race effect on credibility illustrates this strength. Second, the inflation of certain effects, such as evaluation and engagement, shows that simulations may **overemphasize salient identity cues**, especially when human variability and context are stripped away. Third, the failure to reproduce communicator-race effects highlights how simulations may miss **subtle, context-dependent processes** like stereotype activation, which likely require human cognition to manifest.

More broadly, the findings illustrate how simulation can be positioned within communication research. Simulated data can be used as exploratory replications, testing the resilience of theoretical effects before human-subjects data are collected. They can serve as theory probes, clarifying which predictions are strong enough to emerge under artificial conditions and which depend on contextual nuance. They can also function as replication triage, helping researchers prioritize which effects deserve costly, large-scale replications [Collaboration, 2015, Camerer et al., 2018].

At the same time, simulation should not be viewed as a substitute for empirical work. Artificial participants lack lived experience and cannot capture the full range of social dynamics present in real interaction [Bisbee et al., 2024]. The present study illustrates how simulations can inflate effect sizes and erase subtle ones [Hofmann et al., 2024]. As such, simulations are best conceptualized as a **complement** to traditional methods. The most promising pathway forward involves hybrid pipelines in which simulation informs experimental design, which is then validated through preregistered studies with human participants.

In conclusion, simulation offers a valuable new tool for social science research. It can accelerate theory testing, facilitate replications, and clarify boundaries of generalizability. But it also introduces distortions that must be acknowledged. Researchers are therefore encouraged to integrate simulation into multi-method strategies, treating it as an exploratory aid rather than definitive evidence. In doing so, the field can take advantage of simulation's efficiency while preserving the richness of human data essential to understanding communication processes.

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
